# Selection of an Efficient Classification Algorithm for Ambient Assisted Living: Supportive Care for Elderly People

**DOI:** 10.3390/healthcare11020256

**Published:** 2023-01-13

**Authors:** Reyadh Alluhaibi, Nawaf Alharbe, Abeer Aljohani, Rabia Emhmed Al Mamlook

**Affiliations:** 1Department of Computer Science, College of Computer Science and Engineering, Taibah University, Madinah 41477, Saudi Arabia; 2Department of Computer Science, Applied College, Taibah University, Madinah 46537, Saudi Arabia; 3Department of Industrial Engineering and Engineering Management, Western Michigan University, Kalamazoo, MI 49008, USA; 4Department of Aeronautical Engineering, Al Zawiya University (Seventh of April University), Al Zawiya City P.O. Box 16418, Libya

**Keywords:** AAL, MCDM, fuzzy TOPSIS, supportive care, elderly people

## Abstract

Ambient Assisted Living (AAL) is a medical surveillance system comprised of connected devices, healthcare sensor systems, wireless communications, computer hardware, and software implementations. AAL could be used for an extensive variety of purposes, comprising preventing, healing, as well as improving the health and wellness of elderly individuals. AAL intends to ensure the wellbeing of elderly persons while also spanning the number of years seniors can remain independent in their preferred surroundings. It also decreases the quantity of family caregivers by giving patients control over their health situations. To avert huge costs as well as possible adverse effects on standard of living, classifiers must be used to distinguish between adopters as well as nonadopters of such innovations. With the development of numerous classification algorithms, selecting the best classifier became a vital and challenging step in technology acceptance. Decision makers must consider several criteria from different domains when selecting the best classifier. Furthermore, it is critical to define the best multicriteria decision-making strategy for modelling technology acceptance. Considering the foregoing, this research reports the incorporation of the multicriteria decision-making (MCDM) method which is founded on the fuzzy method for order of preference by similarity to ideal solution (TOPSIS) to identify the top classifier for continuing toward supporting AAL implementation research. The results indicate that the classification algorithm KNN is the preferred technique among the collection of different classification algorithms for the ambient assisted living system.

## 1. Introduction

The population is getting older all over the world. This causes several issues for the societies involved. There, a larger proportion of the elderly must be cared for by a smaller proportion of the adolescent group. The utilization of technology is seen as a remedy for continuing to provide adequate care. Ambient assisted living (AAL) is the utilization of devices and systems to keep older people in their homes safe as well as allowed to age in place. Intelligent technologies, wireless networks, software applications, computers, as well as healthcare sensors are all part of it. Such technologies enable people to age in place. AAL allows people’s lives convenient as well as, to a certain extent, self-sufficient [1,2,3,4,5]. The tools and approaches being used in AAL are user-centered as well as incorporated into the individual people’s living environments. It is a massive benefit for elderly individuals because it aids in preventative measures, treatment, as well as the enhancement of their health and wellbeing. AAL techniques are supposed to overcome troubles imposed by recent global socioeconomic advancements. At first, the worldwide population’s ageing process can be seen there. In this case, low birth rates contrast with a higher proportion of elderly citizens with long lifespans. At the very same time, a decrease in the supply of healthcare services leads to an increase in the cost of care. Furthermore, the general rise in chronic illnesses, as well as the desire of the aging population or the disabled to remain in their familiar surroundings, places strain on several nations’ latest medical systems. As a result, information and communication systems have been studied and are expected to handle or mitigate the existing challenges. The dearth of formal primary caregivers can be reduced, particularly in the family environment in which AAL alternatives are used. Additionally, because circumstances may be diagnosed early as well as, thus, healthcare expenditure reduced, visits to doctors may be reduced. The global developments put additional strain on medical systems; however, the use of AAL technology is being researched to alleviate these developments [6,7,8,9,10].

The significant rise in the population’s life expectancy results in a greater number of older people than any other age category. Whereas the extra years are generally spent with severe disabilities, the requirement for caregiving, residential assistance, rehabilitative services, as well as physical support raises the costs of healthcare in developing countries. For instance, healthcare prices in the United States increased to US$3.6 trillion in 2019, necessitating the provision of appropriate assistance systems to enhance elderly people’s standard of living as well as facilitate them in living extremely productively while ageing at a reasonable cost. To that end, the AAL area of research is aggressively growing, and massive projects have already been launched to facilitate the growth of novel AAL systems, with the goal of providing multiple functionalities to the intended audience. This is further evidence of a strong industry in the studied area [11].

Software companies as well as research organizations are working to produce, combine, analyze, and efficiently use big data from diverse as well as distributed source materials to aid at a residence. According to research conducted in the United States, there is increased interest in catching the advantages of utilizing big data based on preconceptions of its influence on the performance and availability of healthcare delivery, as well as in identifying diseases at earlier phases to be handled most effectively. There is also an assumption of more quickly and effectively managing health populations as well as individual people, as well as detecting corruption cases. Big data is anticipated to fundamentally convert smart homes as well as AAL service distribution in the anticipated period, as well as supervisory and economic facets of healthcare delivery, marketing strategies, and governance practices.

Patients over the age of 65 are more likely to be ill. Some healthcare facilities now have specialized elderly population emergency departments (EDs) attended with geriatric-trained physicians and nurses that might contribute to a decrease in hospital admissions. Additionally, over half of all patients are given new medications. Older individuals may visit the ED as a replacement for the general practice or because their primary care physician is not providing adequate care. ED consultations are frequently precipitated by a collapse in an especially elderly patient’s family system; for example, the absence or disease of their caretaker may lead to individuals contacting an ambulance instead of heading to their doctor’s office. Throughout many cases, moreover, the reasons for arriving are genuine emergency situations. Figure 1 shows a graphical demonstration of an AAL system environment.

Ambient assisted living as well as ambient intellectual capacity has had a significant impact, particularly in the past few decades. It is primarily due to an older population and individuals suffering from cognitive illnesses. The worldwide population is ageing faster, as well as the rate being much higher in some regions around the world. As per the UN, Europe has the maximum percentage of the ageing population in comparison to the total citizenry, accompanied by North America. Since the latest COVID-19 pandemic, the medical industry has been under heavy pressure to manage pandemic-infected patients while also monitoring the routine checks as well as tracking of aged people and sensitive patients. With inpatient facilities becoming a health threat for the aging population, who are more susceptible to infection as well as have low resistance, the supply for elderly care beyond medical centers is growing. Numerous technologies have been established to create the utilization of assistive technology that is more acceptable as well as convenient for elderly people, with the goal of reducing or even replacing human help and support. Nevertheless, there are several numbers of challenges, particularly in the engagement of the elderly with assistive processes [11].

AAL contributes to a reduction in hospital service cost, thereby improving the elderly’s living conditions. Numerous different sensors are incorporated in the AAL environment to collect a wide range of information. In the AAL environment, diagnosing and developing the activities of daily living (ADL) dataset is difficult. We should first understand fully what operations the user is performing, how they operate, and how they advance. The goal of ADL controlling is to distinguish between medical issues and those caused by insufficient exercise. The primary goal of AAL is to increase the life survival rate of the elderly in their preferred environment by utilizing personalized health-monitoring equipment as well as information and communication technologies (ICTs) [12,13,14]. AAL encourages research into more flexible living standards, which are evolving into more constructive ways of ageing, alongside looking at just how care is delivered. Furthermore, AAL will be used as a result of innovation to motivate the most recent servicing approaches. It may be thought-provoking research; moreover, the biggest hurdle will be searching for numerous ADLs as well as self-categorization. This paper describes the use of the multicriteria decision-making (MCDM) method, which is based on the fuzzy technique for order of preference by similarity to ideal solution (TOPSIS), to identify the best classifier for further AAL implementation research. Numerous characteristics are used in the representation of information in an obtained dataset. Some of them, however, may be associated with the main objective concept.

The remaining research is divided into several sections. The theoretical purpose of this research is presented in Section 2, which is centered on a literature review of previous research on classification algorithms for ambient assisted living. The fuzzy-TOPSIS approach, as well as the linguistic characteristics and criteria used in this research study, are described in Section 3. A statistical finding is presented in Section 4 to test and evaluate model as well as research findings. Section 5 discusses the study’s significance. Ultimately, Section 6 summarizes the findings, and areas for future research.

## 2. Literature Review

AAL is a newer method of information and communication that introduces solutions as well as recognizes numerous products that allow the elderly and disabled to remain independent as well as enhance their standard of living. It also helps to reduce the cost of healthcare resources. Monitoring systems and devices are installed in the AAL surroundings to collect a wide range of data. Furthermore, AAL will serve as the motivating technology for the most recent. In a rapidly changing world, this could be thought-provoking investigations, and exploring and evaluating various classification algorithms is going to be significant challenge. This section explores previous proposed work regarding information detection or human activity, recognition based on feature selection, cluster analysis, as well as classification strategies.

Oguntala et al. [15] proposed an ambient human activity classification framework which augments data from passive RFID tags’ obtained signal strength indicators (RSSI) to collect accurate activity characterization. To simulate the study objectives, key indices of role, alignment, movement, as well as degree of operations were used with four volunteers to direct reliable clinical managerial decisions. A densely integrated series long short-term memory recurrent neural network model (LSTM RNN) with two layers was incorporated. Using SoftMax, the LSTM RNN prototype retrieves the RSS feature from sensor information as well as classifies the selected sample operations. The effectiveness of the LSTM method was assessed for various data sizes, and the RNN hyperparameters were adapted to optimized states, yielding a precision of 98.18%. Their proposed framework is well-suited for smart homes as well as smart wellness, and it provides a widespread sensing setting for older adults, people with disabilities, and people suffering from chronic disease.

Hassan et al. [16] proposed an integrated AAL framework (HAAL-NBFA) using the naive Bayes-firefly procedure for tracking older patients with prolonged illnesses. This structure takes advantage of current IoT advancements by incorporating ambient along with biomedical sensor-built systems to gather information from old-aged patients as well as then coalesce it into context nations to anticipate the patient’s condition data in real time utilizing context-awareness methodologies. To deal with large imbalanced data arising from the long-term tracking of older patients, the suggested HAAL-NBFA approach suggests a five-phase classification method. The firefly algorithm (FA) was used in their work to optimize the naive Bayes classifier (NB), that also chooses the fewest features with the maximum accuracy. In the event of a sensor failing, their suggested NB-FA performs as a safe-fail module, determining when to halt the system as well as when to allow it to continue.

Belmonte-Fernández et al. [17] introduced a Wi-Fi fingerprinting-based indoor positioning mechanism for wearable technology. Wearable smartwatches were utilized to collect Wi-Fi power signals from nearby wireless access points, which were then used to create a combination of machine learning (ML) classification procedures. The collective algorithm, once created, was utilized to discover a user based on Wi-Fi power signals supplied through the wearable sensors. The investigational findings for five distinct city apartments demonstrated that the scheme was efficient and reliable enough to locate a client at the room stage of development into his or her residence. Another intriguing feature of their proposed system was that it does not necessitate the implementation of any facilities as well as is inconspicuous, as the only gadget needed for it to function is a smart watch.

Zdravevski et al. [18] presented a generic feature-engineering process of producing reliable classification techniques by choosing robust features from various sensors. A wide range of times as well as frequency directory features were derived from the initially captured time series as well as some freshly created time series, i.e., orders of magnitude, first derivative products, delta sequences, as well as fast Fourier transformation (FFT)-centered chain. The number of produced features was, therefore, drastically reduced utilizing two-phase feature selection. Ultimately, various classification approaches were trained as well as tested on a separate trial set. Their suggested procedure was tested on five existing datasets, and it outperformed hand-tailored features in all of them.

Eisa and Moreira [19] introduced an algorithm to handle sensor data flows as well as evaluated sensor-driven characteristics which define the everyday mobility habit of the older adults as a component of the advanced behavior-monitoring system (BMS). They achieved poor detection latency with an affirmation period that was long enough to communicate the finding of several shared emergency circumstances. They also demonstrated as well as assessed the BMS using synthetic information recorded by a developed data transformer intended to mimic various pieces of user mobility profile information at home, as well as a real-life dataset gathered from existing research work. According to their findings, the BMS automatically detects numerous mobility adjustments that can be symptomatic of common health issues.

Bourke et al. [20] characterized the development and implementation of algorithms for accident detection, activity categorization, as well as energy spendings in a remote monitoring structure. Such procedures were developed and verified in a 28-day end-user trial entailing nine elderly participants.

Syed et al. [21] established a novel medical-monitoring framework for AAL that uses IoMT as well as smart machine learning algorithms to supervise the physical exercise of aged persons for faster assessment, decision-making process, as well as improved treatment suggestions. Information is stored from switching function sensors positioned on the patient’s right forearm, left ankle, as well as chest and transferred to the interconnected cloud and database management layer via IoMT gadgets. Hadoop MapReduce strategies were utilized to practice massive volumes of data in comparison. When trying to make a comparison to serial processing, the multinomial naive Bayes classifier, that was also set into the MapReduce approach, was used to recognize the indication encountered through various body portions as well as offered higher adaptability and improved performance with parallelization.

Gulati and Kaur [22] proposed a fuzzy-argument-centered classification arrangement called classification enhanced with fuzzy argumentation (CleFAR). Their anticipated method is used in fall-prevention application fields to categorize fall actions. A new framework for an effective deterrence system based on fall activity recognition (FAR) was introduced. The suggested scheme is intended for decreasing action identification in intelligent-home AAL methods. To assess the scheme’s effectiveness experimentally, intelligent-home AAL surroundings were simulated, and the inhabitant’s regular operation dataset was produced. Wearable drop-detection systems were used to simulate the drop actions.

Zaric et al. [23] initiated research actions to create a computerized decision algorithm for detecting various water-flows discovered in a typical bathroom. They also presented preliminary findings. Time–frequency characterizations were used to analyze as well as characterize various water-flow messages to deliver input variables to the algorithm. Such methods and algorithms can be utilized in a variety of uses that necessitate computerized decision making.

Patel and Shah [24] investigated human activity acknowledgement as a time-series classification task. With the help of the same dataset, nine machine learning as well as deep learning algorithms were implemented and tested. Various parameters were also used to analyze the findings. The purpose of their work was to support in the selection of a pragmatic machine learning strategy for the action recognition system in ambient assisted living processes. According to the comparative study, the long short-term memory (LSTM) network showed the best performance in deep learning, with classification results of 92%.

## 3. Materials and Methods

MCDM is a methodical approach to choosing the most suitable option from a set of viable options. Most actual complications experienced in industry sectors, healthcare facilities, and tertiary institutions, among others, present decision makers with the challenging task of selecting the best option from among numerous options while taking into account multiple factors. TOPSIS and AHP are two MCDM tools that can be employed to tackle issues of this type. Moreover, each of the MCDM techniques has one or more limitations. In the implementation of the analytic hierarchy process (AHP) method, for instance, as the number of participants as well as criteria grows, this increases the difficulty of the decision-making procedure. The MCDM tool’s application is also complicated by the ambiguity of human judgments that also makes it challenging for decision-makers to allocate a precise quantitative score. To tackle the problem of the ambiguity of human judgments, the utilization of fuzzy set theory (FST) in combination with MCDM methods such as TOPSIS has become essential, because the fuzzy system enables the use of linguistic terms that decision makers are much more comfortable with. This section discusses the different materials and methods used in the selection of an efficient classification algorithm for ambient assisted living.

### 3.1. Hierarchy for the Selection Procedure

The necessities and guidelines of the various classification algorithms must be considered when defining the decision criteria for selecting an efficient classification algorithm for AAL. These criteria must consider the performance of classification algorithms as well as the operations and maintenance uncertainty associated with the service being provided. MCDM has emerged as a branch of operations investigation that focuses on creating computational as well as numerical-method tools to facilitate decision makers’ qualitative assessment of performance standards. Numerous studies have been conducted with the goal of creating MCDM. The fuzzy TOPSIS methodology is a MCDM tool with different constraints. A common practice in the TOPSIS procedure is to evaluate solely based on the interviewee’s views and experiences, without needing any statistics. Moreover, it is recommended that when using the FTOPSIS procedure, predetermined linguistic terms are associated with qualitative as well as quantitative scales to continue driving the efficiency assessment of each set of criteria. A total of seven criteria were considered, as listed below in Figure 2 and Table 1.

#### 3.1.1. Decision Tree (DT)

A decision tree is a form of supervised ML that is utilized to categorize or forecast based on the answers to a prior series of questionnaires. The approach is supervised learning in the sense that it is trained as well as tested on data containing the preferred classification. The decision tree does not always offer a direct answer or judgement. It may rather show possibilities so that the data analyst can create an informed judgement under their own. Because decision trees mimic human thoughts, data scientists can perfectly explain and interpret the findings.

#### 3.1.2. Neural Networks (NN)

Neural networks are a collection of algorithms which are weakly modelled after the human brain and are intended to recognize patterns. They perceive sensory data by labelling or grouping raw inputs using a machine perspective [25,26,27]. They recognize numerical patterns enclosed in vectors, into which all real-world information, whether images, sound, message, or time series, should be transformed. Researchers can use neural networks to cluster as well as classify data. They should be considered a clustering as well as classification layer atop the data you collect and retrieve. They aid in the grouping of unlabeled data based on similarities between example inputs, and they also they categorize information once they have a dataset to train on.

#### 3.1.3. Classification and Regression Tree (CART)

A CART is a forecasting model that describes how the values of an outcome measure can be estimated based on some other values. A CART outcome is a decision tree, with each fork representing a split in a predictor variable as well as each end node representing a forecasting for the outcome measure. The CART procedure is a classification method that is utilized to build a decision tree created on Gini’s impurity index. This is a basic machine learning method with a wide range of applications. Leo Breiman, a statistician, invented the term to define decision tree algorithms that can be utilized for classification and regression predictive modelling problems.

#### 3.1.4. Support Vector Machine (SVM)

An SVM is a deep learning algorithm which uses supervised learning to classify or predict data clusters. Supervised learning systems in AI as well as machine learning both provide input as well as preferred output data, that are labelled for classification. This same classification serves as a foundation for future information processing. Support vector machines are used to arrange two datasets based on similarity. The algorithms use lines (hyperplanes) to divide the groups based on patterns. An SVM creates a learning prototype that allocates training examples to one of two groups. SVMs are referred to as non-probabilistic, Boolean linear classifiers by these operations. SVMs could use techniques such as Platt scaling in probabilistic classification configurations.

#### 3.1.5. K-Nearest Neighbor (KNN) Algorithm

A KNN is an information classification method that quantifies how probable a data point is to belong to one of two groups centered on which data points that are closest to it. The k-nearest-neighbor algorithm is an instance of a “lazy learner,” which means that it does not develop a design using the training dataset until the dataset is queried.

#### 3.1.6. Naïve Bayes (NB)

The naive Bayes procedure is a supervised learning procedure that uses the Bayes concept to solve classification complications. It is primarily utilized in text classification with a large training data source. The naive Bayes classifier is a simple as well as well-organized classification algorithm that supports in the development of quick ML models skilled in building quick forecasts. It is a classification algorithm, which implies it predicts based on an object’s likelihood.

### 3.2. Fuzzy TOPSIS Approach

The TOPSIS approach is an MCDM technique formed by two researchers, Hwang and Yoon, that is centered on the “comparative nearness to an optimal situation” concept. In other statements, the main objective is to choose a consequence that is as close to the positive ideal solution (PIS) as imaginable and as far away from the negative ideal solution (NIS) as is practical from a set of available alternatives [28,29,30]. After specifying the weights for every predefined criterion, the results are computed and normalized, and the geometric distance between every alternative as well as the PIS and NIS is assessed [31].

The top option is chosen using the closeness coefficient that may be thought of as a reinforced categorization index that indicates which alternative is contiguous to the ideal solution [32]. As previously stated, the input data that make up the decision matrix in the traditional TOPSIS strategy must be quantitative in nature and well-described [33]. Notwithstanding this being a simple attempt to analyze, its application in isolation is ineffective in attempting to solve selection problems because it cannot manage uncertainty. However, by incorporating fuzzy logic into the TOPSIS process, this may be incorporated to interact through the ambiguity of evaluation data. Traditional TOPSIS is combined with the fuzzy set concept in fuzzy TOPSIS, in which weights are described as linguistic phrases as well as transformed to fuzzy ideals. It is described as a multicriteria decision-making instrument [34].

As shown in Figure 3, a stage process sequential approach for weighting computation as well as significance scoring using fuzzy (TOPSIS) is as follows:

**Stage 1:** Construct a decision matrix.

The fuzzy (TOPSIS) strategy is used to evaluate five criteria as well as six alternatives in this research study. The term “category” refers to the various types of criteria. Assume the decision-making group has K members. If the kth evaluation specialist’s fuzzy record and priority weight for the ith alternative on the jth criterion are:(1)$${\overset{ˇ}{x}}_{ij}^{k}=\left(a_{ij}^{k},b_{ij}^{k},c_{ij}^{k}\right)\;$$
(2)$$\mathrm{and}\;{\overset{ˇ}{w}}_{j}^{k}=\left(w_{j1}^{k},w_{j2}^{k},w_{j3}^{k}\right)$$
where i = 1, 2, 3, …, m, as well as j = 1, 2, 3, …, n, then the combined fuzzy rankings ${\overset{ˇ}{x}}_{ij}$ of options $\left(i\right)$ with reference to every condition $\left(j\right)$ are quantified with the help of
(3)$${\overset{ˇ}{x}}_{ij}=(a_{ij},b_{ij},c_{ij})$$

Table 2 demonstrates the category of condition as well as the weight applied to each condition. The symbol for a triangular fuzzy amount (TFN) is (L, M, U). The characteristics L, M, as well as U, respectively, represent the least effective, most effective, as well as maximum possible values. The fuzzy measure utilized in the model is depicted in Table 3 below.

**Stage 2:** Generate the normalized judgment matrix.

Additionally, a normalized judgment matrix ${\tilde{r}}_{ij}$ might be determined through using the subsequent relation dependent on the positive as well as negative ideal possibilities:(4)$${\tilde{r}}_{ij}=\left(\frac{a_{ij}}{c_{j}^{*}},\frac{b_{ij}}{c_{j}^{*}},\frac{c_{ij}}{c_{j}^{*}}\right)\;\;c_{j}^{*}=max_{i}\;c_{ij};\;Positive\;ideal\;solution$$
(5)$${\tilde{r}}_{ij}=\left(\frac{a_{j}^{-}}{c_{ij}},\frac{a_{j}^{-}}{b_{ij}},\frac{a_{j}^{-}}{a_{ij}}\right)\;\;a_{j}^{-}=min_{i}\;a_{ij};\;Positive\;ideal\;solution$$

**Stage 3:** Produce the weighted normalized judgment matrix.

The weighted normalized judgment matrix may be generated by increasing the weight of every measure in the standardized fuzzy decision matrix using the equation as follows, weighing each criterion according to its relative importance.
(6)$${\tilde{v}}_{ij}={\tilde{r}}_{ij}.{\tilde{w}}_{ij}\;$$
where ${\tilde{v}}_{ij}$ denotes the weighted normalized judgment matrix and ${\tilde{w}}_{ij}$ denotes the weight of criterion $c_{j}$.

**Stage 4:** Calculate both the fuzzy negative perfect alternative ($FNIS,A^{-}$) and the fuzzy positive ideal alternative (*FPIS, A**).

Equation (7) as well as Equation (8) can be used to describe the *FPIS* as well as the *FNIS* of the alternative solutions, respectively:(7)$$A^{*}=\left\{{\tilde{v}}_{1}^{*},{\tilde{v}}_{2}^{*},\ldots ,{\tilde{v}}_{n}^{*}\right\}=\left\{\left(\underset{j}{max}v_{ij}|i\in B\right),\left(\underset{j}{min}v_{ij}|i\in C\right)\right\}\;$$
(8)$$A^{-}=\left\{{\tilde{v}}_{1}^{-},{\tilde{v}}_{2}^{-},\ldots ,{\tilde{v}}_{n}^{-}\right\}=\left\{\left(\underset{j}{min}v_{ij}|i\in B\right),\left(\underset{j}{max}v_{ij}|i\in C\right)\right\}\;$$

Here, ${\tilde{v}}_{i}^{*}$ is the maximum amount of *i* for all the alternative solutions and ${\tilde{v}}_{1}^{-}$ is the minimum extent of *i* for all the alternative solutions. *B* and *C* symbolize the positive as well as negative perfect solutions, likewise.

**Stage 5:** Determine the gap among every alternative as well as the fuzzy positive ideal solution *A**, as good as the distance amongst each interim answer and as good as the fuzzy negative ideal result $A^{-}$.

The gap amongst each alternative solution as well as *FPIS* and among every alternative as well as *FNIS* is computed through the subsequent Equations (9) and (10), respectively:(9)$$S_{i}^{*}=\sum_{j=1}^{n}d({\tilde{v}}_{ij},{\tilde{v}}_{j}^{*})i=1,2,\ldots ,m$$
(10)$$S_{i}^{-}=\sum_{j=1}^{n}d({\tilde{v}}_{ij},{\tilde{v}}_{j}^{-})i=1,2,\ldots ,m$$

*d* is the range in the middle of two fuzzy numerals, when supposed two triangular fuzzy figures ($a_{1},b_{1},c_{1}$) as well as ($a_{2},b_{2},c_{2}$); the e range among the two may be estimated as following Equation (11):(11)$$d_{v}\left({\tilde{M}}_{1},{\tilde{M}}_{2}\right)=\sqrt{\frac{1}{3}\left[{\left(a_{1}-a_{2}\right)}^{2}+{\left(b_{1}-b_{2}\right)}^{2}+{\left(c_{1}-c_{2}\right)}^{2}\right]}\;$$

Note that $d\left({\tilde{v}}_{ij},{\tilde{v}}_{j}^{*}\right)$ and $d\left({\tilde{v}}_{ij},{\tilde{v}}_{j}^{-}\right)$ are crisp figures.

**Stage 6:** Directly measure the proximity coefficient as well as prioritize the choices.

The following expression can be used to measure the proximity coefficient of each alternative:(12)$$CC_{i}=\frac{S_{i}^{-}}{S_{i}^{+}+S_{i}^{-}}\;$$

There are many different tools for assessing and classifying options, each of which has a variety of constraints. Every strategic approach has its own set of pluses and minuses. The fuzzy TOPSIS process has the merits of being simple in its statistical model, convenient to depict human attitudes, as well as designed to allow for clear and unambiguous trade among multicriteria. Moreover, the technique is characterized as a revealing theory, with the conception that even though there is no optimal solution, a strategy with optimized attributes on all characteristics can be observed. As a direct consequence, fuzzy TOPSIS including a trapezoidal membership estimation is utilized to determine the basic competences of various classification algorithms for ambient assisted living in this study.

## 4. Results

The decision makers’ point of view was not given equal weight in the procedure for making decisions for the selection of an efficient classification algorithm for ambient assisted living. The team of decision makers involved 50 from the software industry with more than ten years of experience, 20 academics with more than twelve years of experience, and 20 researchers with more than eight years of experience. As a result, a weight variable is initiated to signify the significance for that criterion in the decision maker’s viewpoint. Each criterion is assigned one of five levels of significance: very low, low, medium, high, and very high. The 90 experts assessed the six competing classification algorithms (DT, NN, CART SVM, KNN and NB), taking into account the decision criteria weights for increased operating quality applicability. The evaluation was based on professional opinion, with each criterion receiving a distinct linguistic significance to represent the classification algorithm’s effectiveness as shown in Figure 4 and Table 4, Table 5, Table 6, Table 7, Table 8 and Table 9.

The CCi figures were ranked so that the maximum preference corresponded to a classification algorithm through a CCi significance nearer to 1, and the lowest priority corresponded to a CCi value nearer to 0. KNN > DT > NN > NB > CART > SVM is the outcome of the fuzzy-TOPSIS rating system. Because the closeness significance level is near to the ideal solution significance (1), the classification algorithm KNN must be regarded as the best approach among the set of classification algorithms for ambient assisted living evaluated in this problem, as shown in Table 9 and Figure 5.

## 5. Discussion

By combining sensors, electronic controls, smart functionalities, as well as artificial intelligence, AAL creates a caring living environment. Conventional technological tools for individuals with disabilities; widespread creative approaches to accessibility, functionality, as well as the appropriateness of technological devices; as well as the evolving ambient intelligence (AI) computing concept, that also provides intelligent, inoffensive, as well as prevalent assistance, are the foundations of AAL. Recent AAL mechanisms offer great possibilities for older individuals to preserve their freedom while also tracking and improving their health and wellbeing. Wearable and implantable sensors, assistive robotic systems, home automation, as well as smart fabrics are just a few of the key technological innovations that have created the opportunity. In the meantime, advances in computational strategies have aided in the full realization of such technologies’ potential. However, there are also significant problems that researchers must address in the future. In the past couple of years, there has been increasing interest in developing AAL solutions that will enable people to live independently. From a social as well as economic standpoint, the adjustment towards the aging population has posed new problems for today’s generation. AAL can provide a number of solutions for improving people’s standard of living, enabling them to live a healthy lifestyle and also more freely for extended periods of time. An AAL system uses various classification algorithms to support the elderly, caregivers, and health specialists.

Classification is a controlled machine learning tactic that allocates labels or classes to objects or groups that differ from one another. Classification consists of two stages. The first stage is model development, which would be characterized as the analyzation of a database’s training materials. The second stage is to apply the designed model to classification. This same proportion of accurately classified specimen or records is used to calculate classification performance. Scientific experiments, clinical diagnosis, weather forecasting, credit monitoring, customer insights, target marketing, as well as criminal identification have all been major applications of classification. Numerous research teams have already been implementing different algorithms to support healthcare professionals with greater precision in the treatment and monitoring of old-aged persons through AAL systems. The selection of an efficient classification algorithm for an ambient assisted living system may be a challenging task. The finding in this research shows that the KNN is measured as the preferred algorithm, among the group of different classification algorithms for efficient AAL system development. Another research work conducted by OrtizBarrios [35] supports our findings in this study. In their research findings, KNN was identified as the best classifier for assisting people with dementia in adopting assistive technology. Several other studies [36,37,38,39] have also demonstrated that the KNN classification is accurate and efficient. The KNN algorithm is indeed very simple to comprehend as well as develop. To classify a fresh piece of data, the KNN algorithm searches the entire dataset for K-nearest neighbors. Many classifier algorithms are simple to incorporate for binary issues as well as require an additional attempt to implement for multiclass challenges, whereas KNN adapts to multiclass challenges with no additional attempt.

## 6. Conclusions

AAL is a digital communication technology that generates a smart object in the setting to help the elderly live comfortably. These days, the number of senior citizens living alone is increasing. As a result, it has a significant impact on our society. Efficient performance is required for the productive implementation of AAL systems, which also guarantees end-user acceptance. This research study presents a fuzzy TOPSIS method for selecting the most efficient classification algorithm for ambient assisted living in order to provide supportive care for elderly people. The classification algorithm KNN, therefore, is considered as the greatest approach among the group of classification techniques for AAL since the closeness level of significance is nearest to the ideal solution significance. Future research in the encompassing research work would then look to see if it is possible to investigate the objective evaluation of the models presented in this paper. Undoubtedly, there is potential to create an insightful collaborative mobile-based analysis platform that would allow service providers to conduct a real-time technology-acceptance-likelihood evaluation for AAL predicated on a small set of input variables. The outcomes of a feasibility analysis would be utilized to improve and supplement the AAL system designs presented in this study.

## Figures and Tables

**Figure 1 healthcare-11-00256-f001:**
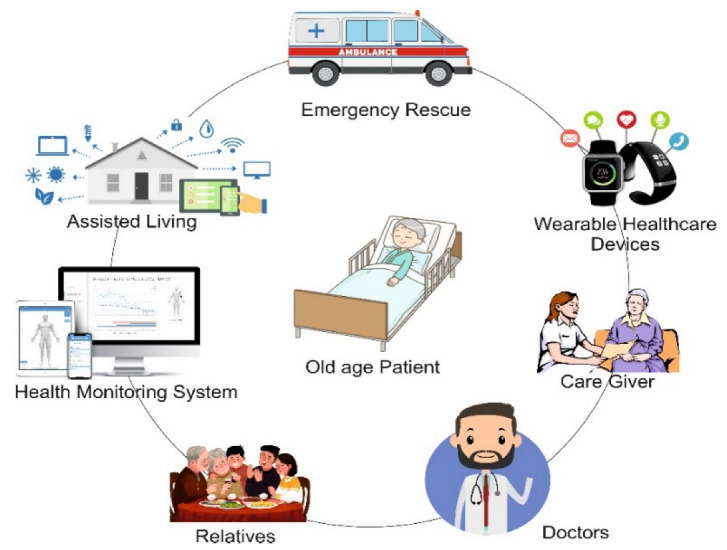
Graphical illustration of an AAL system environment.

**Figure 2 healthcare-11-00256-f002:**
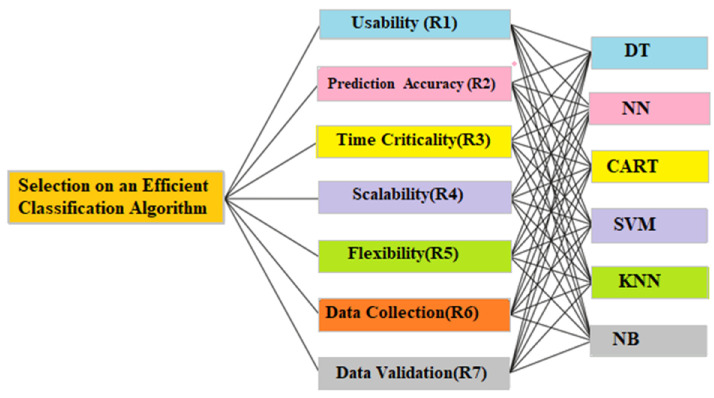
Hierarchical structure for the MCDM assessment.

**Figure 3 healthcare-11-00256-f003:**
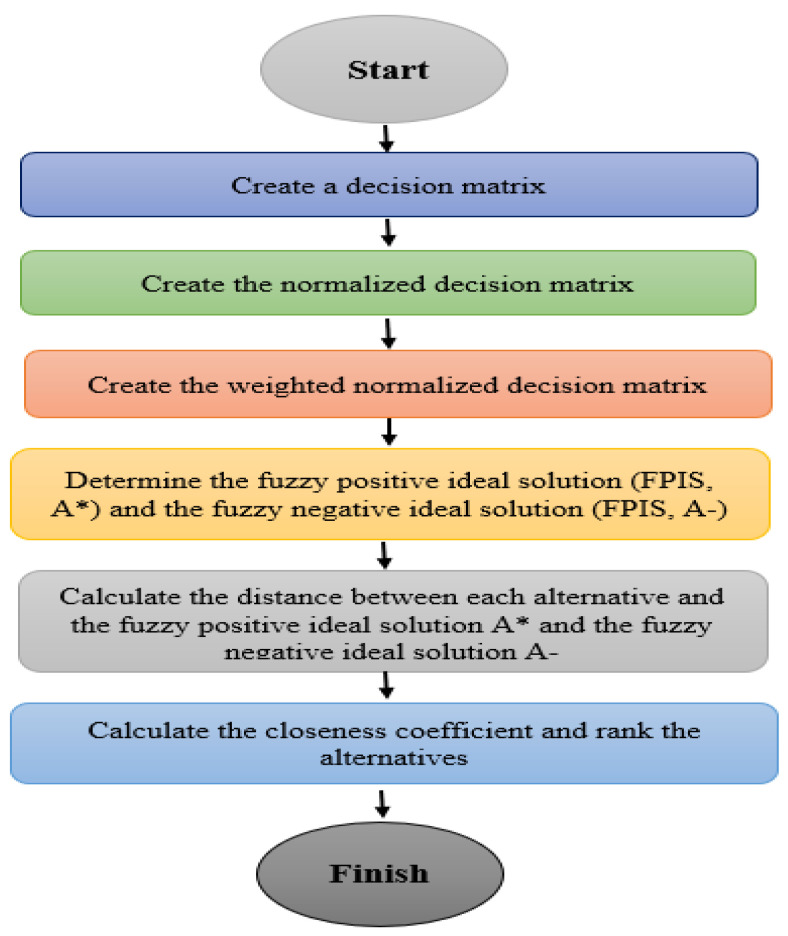
Flow diagram of Fuzzy TOPSIS approach.

**Figure 4 healthcare-11-00256-f004:**
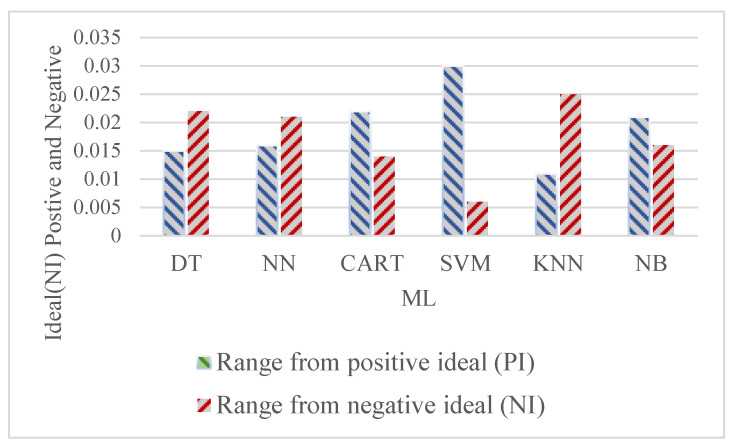
Range from positive as well as negative ideal alternative solutions.

**Figure 5 healthcare-11-00256-f005:**
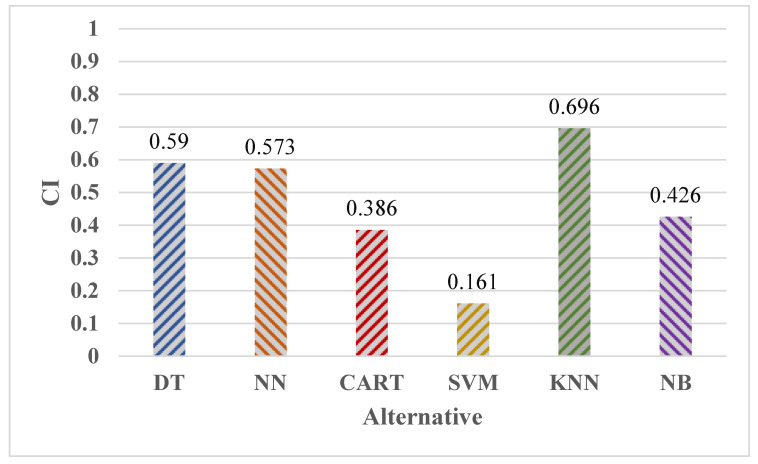
Closeness coefficient of each alternative.

**Table 1 healthcare-11-00256-t001:** Different identified evaluation criteria with their description.

Criteria	Description
Usability (R1)	Usability is a way of measuring of how effectively, competently, as well as sufficiently a particular user in a particular circumstance can utilize a product/design to accomplish a clear objective.
Prediction accuracy (R2)	The accuracy of prediction specifies whether the forecasted values represent the actual values of the intended domain inside the margin of error caused by statistical fluctuations as well as noise in the given input readings.
Time criticality (R3)	The significance of executing the task is demonstrated by the term “time critical.” If a task is highly time sensitive, the contenders are also contemplating it.
Scalability (R4)	Scalability is the degree to which a system can modify its effectiveness as well as expense in reaction to variations in application software computational requirements.
Flexibility (R5)	The easiness with which the process can react to uncertainty in order to maintain or enhance its significance delivery is referred to as flexibility.
Data collection (R6)	Data collection is the comprehensive gathering of observational data or test results.
Data validation (R7)	Before using, producing, or otherwise analyzing information, it is necessary to validate it to ensure its integrity and precision.

**Table 2 healthcare-11-00256-t002:** Features of Different Criteria.

Name	Weight
R1	0.143, 0.143, 0.143
R2	0.143, 0.143, 0.143
R3	0.143, 0.143, 0.143
R4	0.143, 0.143, 0.143
R5	0.143, 0.143, 0.143
R6	0.143, 0.143, 0.143
R7	0.143, 0.143, 0.143

**Table 3 healthcare-11-00256-t003:** Fuzzy-logic-based scale.

Code	Linguistic Terms	L	M	U
1	Very low	1	1	3
2	Low	1	3	5
3	Medium	3	5	7
4	High	5	7	9
5	Very high	7	9	9

**Table 4 healthcare-11-00256-t004:** Fuzzy-based Decision Matrix.

	R1	R2	R3	R4	R5	R6	R7
DT	(4.326, 6.326, 8.281)	(4.393, 6.393, 8.056)	(4.101, 6.101, 7.989)	(4.461, 6.461, 8.034)	(4.213, 6.213, 8.101)	(4.348, 6.348, 8.146)	(4.371, 6.371, 8.169)
NN	(4.348, 6.348, 8.124)	(4.303, 6.303, 8.101)	(4.393, 6.393, 8.101)	(4.371, 6.371, 8.191)	(4.258, 6.258, 7.944)	(4.236, 6.236, 8.056)	(4.371, 6.371, 8.011)
CART	(4.213, 6.213, 8.101)	(4.551, 6.551, 8.236)	(4.079, 6.079, 7.944)	(4.438, 6.438, 8.079)	(4.124, 6.124, 7.899)	(4.281, 6.281, 8.011)	(4.146, 6.146, 7.921)
SVM	(4.326, 6.326, 8.169)	(4.303, 6.303, 8.056)	(4.281, 6.281, 8.079)	(4.079, 6.079, 7.944)	(3.989, 5.989, 7.854)	(4.124, 6.124, 7.944)	(4.169, 6.169, 7.944)
KNN	(4.213, 6.213, 8.079)	(4.506, 6.506, 8.213)	(4.528, 6.528, 8.258)	(4.348, 6.348, 8.169)	(4.348, 6.348, 8.169)	(4.258, 6.258, 8.213)	(4.169, 6.169, 7.966)
NB	(4.438, 6.438, 8.281)	(4.326, 6.326, 8.169)	(4.146, 6.146, 8.034)	(4.326, 6.326, 8.146)	(4.146, 6.146, 8.079)	(4.101, 6.101, 7.966)	(4.191, 6.191, 8.101)

**Table 5 healthcare-11-00256-t005:** A normalized judgment matrix.

	R1	R2	R3	R4	R5	R6	R7
DT	(0.522, 0.764, 1.000)	(0.533, 0.776, 0.978)	(0.497, 0.739, 0.967)	(0.545, 0.789, 0.981)	(0.516, 0.761, 0.992)	(0.529, 0.773, 0.992)	(0.535, 0.780, 1.000)
NN	(0.525, 0.767, 0.981)	(0.522, 0.765, 0.984)	(0.532, 0.774, 0.981)	(0.534, 0.778, 1.000)	(0.521, 0.766, 0.972)	(0.516, 0.759, 0.981)	(0.535, 0.780, 0.981)
CART	(0.509, 0.750, 0.978)	(0.553, 0.795, 1.000)	(0.494, 0.736, 0.962)	(0.542, 0.786, 0.986)	(0.505, 0.750, 0.967)	(0.521, 0.765, 0.975)	(0.508, 0.752, 0.970)
SVM	(0.522, 0.764, 0.986)	(0.522, 0.765, 0.978)	(0.518, 0.761, 0.978)	(0.498, 0.742, 0.970)	(0.488, 0.733, 0.961)	(0.502, 0.746, 0.967)	(0.510, 0.755, 0.972)
KNN	(0.509, 0.750, 0.976)	(0.547, 0.790, 0.997)	(0.548, 0.791, 1.000)	(0.531, 0.775, 0.997)	(0.532, 0.777, 1.000)	(0.518, 0.762, 1.000)	(0.510, 0.755, 0.975)
NB	(0.536, 0.777, 1.000)	(0.525, 0.768, 0.992)	(0.502, 0.744, 0.973)	(0.528, 0.772, 0.995)	(0.508, 0.752, 0.989)	(0.499, 0.743, 0.970)	(0.513, 0.758, 0.992)

**Table 6 healthcare-11-00256-t006:** The weighted standardized judgment matrix.

	R1	R2	R3	R4	R5	R6	R7
DT	(0.509, 0.754, 0.976)	(0.076, 0.111, 0.140)	(0.071, 0.106, 0.138)	(0.078, 0.0113, 0.140)	(0.074, 0.109, 0.142)	(0.075, 0.111, 0.142)	(0.077, 0.112, 0.143)
NN	(0.075, 0.110, 0.140)	(0.075, 0.109, 0.141)	(0.076, 0.111, 0.140)	(0.076, 0.111, 0.143)	(0.075, 0.110, 0.139)	(0.074, 0.109, 0.140)	(0.077, 0.112, 0.140)
CART	(0.073, 0.107, 0.140)	(0.079, 0.114, 0.143)	(0.071, 0.105, 0.138)	(0.077, 0.112, 0.141)	(0.072, 0.107, 0.138)	(0.075, 0.109, 0.139)	(0.073, 0.080, 0.139)
SVM	(0.075, 0.109, 0.141)	(0.075, 0.109, 0.140)	(0.074, 0.109, 0.141)	(0.071, 0.106, 0.139)	(0.070, 0.105, 0.137)	(0.072, 0.107, 0.138)	(0.073, 0.108, 0.139)
KNN	(0.073, 0.107, 140)	(0.078, 0.113, 143)	(0.078, 0.113, 143)	(0.076, 0.111, 143)	(0.076, 0.111, 143)	(0.074, 0.109, 143)	(0.073, 0.108, 139)
NB	(0.077, 0.111, 143)	(0.075, 0.101, 142)	(0.072, 0.106, 139)	(0.077, 0.110, 142)	(0.073, 0.108, 141)	(0.071, 0.106, 139)	(0.073, 0.108, 142)

**Table 7 healthcare-11-00256-t007:** The positive as well as negative ideal resolutions.

	Positive Ideal (PI)	Negative Ideal (NI)
R1	0.077, 0.111, 0.143	0.073, 0.107, 0.140
R2	0.079, 0.114, 0.143	0.075, 0.109, 0.140
R3	0.078, 0.113, 0.143	0.071, 0.105, 0.138
R4	0.078, 0.113, 0.143	0.071, 0.106, 0.139
R5	0.076, 0.111, 0.143	0.070, 0.105, 0.137
R6	0.076, 0.111, 0.143	0.071, 0.106, 0.138
R7	0.077, 0.112, 0.143	0.073, 0.108, 0.139

**Table 8 healthcare-11-00256-t008:** Range from positive as well as negative ideal alternative solutions.

ML	Range from Positive Ideal (PI)	Range from Negative Ideal (NI)
DT	0.015	0.022
NN	0.016	0.021
CART	0.022	0.014
SVM	0.03	0.006
KNN	0.011	0.025
NB	0.021	0.016

**Table 9 healthcare-11-00256-t009:** Summary of Closeness coefficient data.

Alternatives	Ci	Rank
DT	0.59	2
NN	0.573	3
CART	0.386	5
SVM	0.161	6
KNN	0.696	1
NB	0.426	4

## Data Availability

Data sharing does not apply to this article as no datasets were generated during the current study.

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
