# Peer review of "Selection of an Efficient Classification Algorithm for Ambient Assisted Living: Supportive Care for Elderly People"

_healthcare, 2023, doi:10.3390/healthcare11020256_

Round 1
Reviewer 1 Report (Previous Reviewer 2)
In this second round, corrections are vague and not practical. Most of the comments are not addressed, and I'm not sure which content has been added to the paper.
I strongly recommend following a clear response pattern which enables reviewers to precisely know what changes have been made. In my case, I can't identify the corrections, only a concrete paragraph in yellow at the end of the fifth section which I don't identify its justification. Then, I don't know what changes correspond with the reviews I provided.
As a result, I recommend including in the letter for reviewers the content added or changed to satisfy feedback. In another way, it is not possible to understand the improvement of the paper.
Author Response
kindly see attached file

Reviewer 2 Report (Previous Reviewer 3)
no more comments, ready to accept for publication.
Author Response
ready to accept for publication.
Reviewer 3 Report (New Reviewer)
This article is a comparison of several algorithms used in AAL. But the author also did not mention the used platform. There are the following suggestions, the authors have to correct.
Referring to Figure 2, The R2 "A ccuracy" has to correct "Accuracy".
In 3.1.5. k-nearest-neighbor algorithm (KNN), the first letter of characters must be capitalized.
r̃ij, as (1) and (2) which is not defined in the article, as well as ṽij. And the Equations 1, 2 and 3 do not appear too.
In 399's, correcting the A- same to Equation 5 as A--.
In 463's, ...., as shown in Table 8 and Figure 4. In which the authors must check Table 8 or Table 9.
Nouns should be consistent in the text, please make sure that the conclusion listed is KNN or K-NN?
Round 2
Reviewer 3 Report (New Reviewer)
The author has explained the problems of the previous version.
This manuscript is a resubmission of an earlier submission. The following is a list of the peer review reports and author responses from that submission.
Round 1
Reviewer 1 Report
Thank you for your work in this very important topic.
The research presented in this article is strong and important. The study presents an original research results. The statistics were performed to a high technical standard. Conclusions are presented in an appropriate fashion and are supported by the data.
In my opinion, the abstract needs improvement. Please indicate the purpose of the research, methods, the most important conclusions.
I really enjoyed reading your work.
Reviewer 2 Report
The submitted paper introduces a novel technique to identify the most appropriate classification algorithm for Ambient Assisted Living. For this, the work proposes a method which defines a set of steps to identify the closeness coefficient for a classification algorithms. Considering this, the most relevant classification methods are applied under this methodology, identifying K-Nearest Neighbors (KNN) as the best option.
In general, the paper introduces a weak topic and a low-impact contribution. The global aspect of the manuscript is messy and even becomes hard to follow at some points.
In the abstract, the main points of the paper are addressed, introducing Ambient Assisted Living concept but describing classification algorithms and machine learning as a second-priority. I think these two points deserve more attention since their analysis and applicability are the main motivation for the article. Also, initials MCDM are mentioned but the meaning is never described.
In the introduction of the work, there are a lot of efforts describing the demography challenges of society to respond to the increasing requirements from older adults and third age population. However, again, artificial intelligence or machine learning lack of any relevance. In the same way, the potential role of classification algorithms is not mentioned or described. As a result, it is difficult to understand well what is being addressed in the work, since a clear goal for the paper is not introduced.
Also in the introduction, there are a lot of stats and objective information but there are not references to any source. It seems that there has been an error in the cites, because the first one appears at literature review as [15], but there is not trace for [1, 2, 3...].
Considering this, I strongly recommend to improve the speech line for the section and highlight the relevant ideas to understand and to immerse the reader to the problems that you are tackle, as well as the solution you bring.
In the case of section 2, the literature review, the section begins without any instruction of the works involved in the state-of-art process. Are these works related to AAL solutions? Or are they related with classification algorithms? Are they focused on studying the effects of aging? To solve these questions are let the reader know about the section, it is mandatory to include some lines to guide about the reviews which have been done. Thus, it is potentially easy to understand and clarify which are the topics involved in the research and how the already existed proposals align with the one you bring. Therefore, it is required to struct the section into a speech which involves the lines that your researching and how the literature is different from what you are doing. In this point of the paper, following what is being presented becomes really diffuse.
Section 3 involves the materials and methodology followed to perform the research. Again, there is not any introduction to the section, addressing the different existing classification algorithms. A brief explanation of the methodology, before any more detail, would make the reading much easier. Then, detailed explanations of the process are provided, explaining the core of the proposal. This part is quite good. There are some parts which require rewriting and TOPSIS initials are mentioned but not described.
Finally, the results are analysed. For this, the paper explains that a set of 90 experts are involved to determine the operability of the solutions. This part requires special and concrete attention since it is not clear how the experiment have taken place. There are not explanations of how the experts have addressed the process or how this is aligned with the AAL purposes. Thus, the results are obscure, becoming hard to understand or replicate. I would recommend to extend the explanations about the followed process and even include an explanatory figure.
In conclusion, I would strongly recommend to the authors to check my humble review and improve the manuscript. I have not any doubt about how the proposal rocks and becomes a relevant advance. However, the purpose and goals are just a leg of researching. The methods, the systematic processes and the transparency become the other leg which is very relevant to share correctly the advances to the scientific community.
Reviewer 3 Report
The major comments are as follows:
-Many undefined terms, e.g., MCDM
-Some of the terms are already defined, but again their full form has been used in different places instead of
acronym.
e.g., Ambient Assisted Living
-Need to fine-tune the texts.
-Even with survey oriented paper, motivations and clear key contributions should be
explicitly stated and delivered.
-Many tables, but they are not well referred to the text.
-Improve results section!!! Need to produce more results.
-Type of the paper is not clear to me! is it tutorial paper or research paper?
-Rewrite abstract and conclusion.
-Improve introduction.
-Many undefined notation.
-Articulation on intereting part is missing.
-The research design can be improved.
-Overall, hard to follow content and structure of the paper. needs significant improvement.